# Increasing Dietary Lysine Impacts Differently Growth Performance of Growing Pigs Sorted by Body Weight

**DOI:** 10.3390/ani10061032

**Published:** 2020-06-13

**Authors:** Pau Aymerich, Carme Soldevila, Jordi Bonet, Josep Gasa, Jaume Coma, David Solà-Oriol

**Affiliations:** 1Vall Companys Group, 25191 Lleida, Spain; csoldevila@vallcompanys.es (C.S.); jbonet@vallcompanys.es (J.B.); jcoma@vallcompanys.es (J.C.); 2Animal Nutrition and Welfare Service, Department of Animal and Food Sciences, Universitat Autònoma de Barcelona, 08193 Bellaterra, Spain; Josep.Gasa@uab.cat (J.G.); David.Sola@uab.cat (D.S.-O.)

**Keywords:** lysine, growing pig, body weight, requirements

## Abstract

**Simple Summary:**

The increasing demand on animal products expected for the next decades requires animal production systems to become more efficient in resource use. Most commercial operations feed all pigs the same feed at a determined time depending on the average BW of the batch, but without considering the variability of the population. However, low body weight (BW) pigs have been related to extra costs as reduced barn utilization, losses due to the poor carcass grading and inefficiency of phase feeding strategies. Some studies have previously hypothesized the need to do different phase feeding strategies to pigs sorted by initial BW. This work aimed to compare the effect of increasing the standardized ileal digestible lysine to net energy ratio (SID Lys:NE) over the performance of growing pigs sorted by initial BW in 3 categories (small, medium, and large). The results showed that small pigs could use more efficiently high SID Lys:NE diets compared to the large pigs during the growing phase (28–63 kg). The conclusions imply positive effects of feeding higher dietary lysine to small pigs to compensate for their reduced feed intake capacity. This strategy might improve growth rate and feed efficiency, without increasing feed costs per kg gain.

**Abstract:**

An experiment was conducted analyzing whether growing pigs classified in different initial body weight categories (BWCAT) have a different response to increasing standardized ileal digestible lysine to net energy ratio (SID Lys:NE), to assess whether light pigs might benefit from being differentially fed. A total of 1170 pigs in pens of 13 were individually weighed, classified in 3 BWCAT (Lp: 32.1 ± 2.8 kg, Mp: 27.5 ± 2.3 kg, and Sp: 23.4 ± 2.9 kg), and afterwards pens were randomly allocated to 5 dietary SID Lys:NE treatments (3.25 to 4.88 g/Mcal) fed over 47 days. Results reported a greater linear improvement of growth and feed efficiency of Sp compared to Lp when increasing SID Lys:NE. Modelling the response to SID Lys:NE using quadratic polynomial models showed that the levels to reach 98% of maximum growth from day 0–47 were 3.67, 3.88, 4.06 g SID Lys/Mcal NE for Lp, Mp, and Sp, respectively. However, due to the overlapping SID Lys:NE confidence intervals at maximum performance, it was not possible to determine if requirements were different between BWCAT. Summarizing, the results suggested that feeding small pigs greater SID Lys:NE than large pigs can improve their performance and increase the efficiency of the overall production system.

## 1. Introduction

Pigs with a low body weight (BW) continue to be a major concern in all-in all-out swine production systems, as they have been associated with a longer time to reach marketing [1,2,3], a greater mortality rate [2,3,4], and the resulting inefficiency of phase feeding strategies [5]. The latter, being a widespread feeding system, aims to mimic the reduction in the optimal concentration of lysine in the diet required for growth with increasing BW [6,7,8]. Practical application of phase feeding programs consists in delivering a specific amount of each feed to the farm, aiming to fulfill the requirements of the average pig. However, as light pigs are known to have a lower feed intake than their heavier mates [9,10,11,12], they might eat a lower amount of the first feeds, and therefore less lysine during the early stages both in the nursery and growing facilities. Even though phase feeding focuses on feeding pigs more precisely [7], the inherent BW variability of swine production systems [13] might represent that the requirements of the lightest pigs might not be fulfilled when applying this strategies [14].

Inconsistent results regarding whether light pigs have a lower feed intake when expressed relative to metabolic BW [9,10,11] suggest that in addition to a lower BW, there might be other factors involved. Besides, contradictory results have been reported regarding whether lightweight pigs deposit more body fat as a consequence of their limited number of muscle fibers [15] or have a greater relative lean tissue content [10,16,17,18,19]. The reduced feed intake might entail that a greater proportion of the ingested energy is retained as protein [10], but the issue might not be so straightforward if, as suggested by some studies, those pigs had an increased lysine catabolism [20,21]. Furthermore, Jones and Patience [22] determined that there was a significant positive correlation between average daily gain (ADG) during nursery and nitrogen digestibility. Similarly, other authors have reported a greater lysine disappearance as a percentage of total intake at low energy intakes [23]. Consequently, most nutritional studies have focused on feeding high density diets, by increasing amino acid or energy concentrations during the nursery phase, as they considered that early interventions might be more effective [12,24,25,26]. However, other studies have also reported positive effects of dietary interventions during the grow-finishing phase [17,27,28], and others reported no advantage [29,30]. Finally, modelling approaches also support the idea that low BW pigs would require a greater lysine concentration in the diets compared to heavier pigs [6,31]. However, the majority of those studies are based on average population growth performance and requirements are calculated using factorial equations [32]. Thus, doubts arise about the use of these models to calculate different requirement for growing pigs within the same population [33]. Finally, practical application of different feeding plans for pigs from the same population would require one of the following strategies: split feeding of pigs [34] or using precision feeding systems [33].

The hypothesis of the present study was that small growing pigs within a batch might require diets with a higher lysine concentration to maximize lysine intake and therefore growth performance compared to their larger mates. Consequently, this study aimed to compare the effects of increasing standardized ileal digestible lysine to net energy ratio (SID Lys:NE) on growth performance among pigs classified in different body weight categories (BWCAT).

## 2. Materials and Methods

All the procedures described in this work followed the EU Directive 2010/63/EU for animal experiments.

### 2.1. Experimental Design and Animals

In this study, the differential effect of SID Lys:NE between BWCAT on growth performance was analyzed in a dose-response trial. The experiment was conducted for 47 days in a commercial-experimental farm from Vall Companys Group (Alcarràs, Lleida), after a 10 day adaptation period. The day of arrival, a total of 1170 growing pigs ((Pietrain × (Landrace × Large white), half boars and half gilts) were grouped in pens of 13 pigs, with a total of 90 non-mixed sex pens, and individually pre-classified in 3 initial BWCAT. Pigs came from a weekly farrowing sow farm, and although were not followed from birth, maximum age difference was 7 days. The first day of the experiment pigs were reclassified, if necessary (e.g., a large pig in a pen of small pigs), as Large (Lp: 32.1 ± 2.8 kg), Medium (Mp: 27.5 ± 2.3 kg) or Small (Sp: 23.4 ± 2.9 kg). Each pen was randomly assigned by BW to one of the 5 treatments (3.25, 3.66, 4.07, 4.47 and 4.88 g SID Lys/Mcal NE), with 6 replicates per treatment and BWCAT, 3 of each sex. Each pen (3 × 3 m) had a half slatted concrete floor, 1 hole wet-dry feeder and an additional nipple waterer on the other side. Ad libitum access to feed and water was ensured during the whole trial. Pigs were individually weighed and monitored using electronic ear tags at the beginning of the trial, at day 26 and at day 47, at the end of the trial. In addition, pen feed intake was measured weekly by knowing the feed on offer and the amount of feed remaining in each trough and corrected if any pig died or was removed from the trial. The day after finishing the experiment, an ultrasound scan (Tecnoscan SF-1 Wi-Fi back fat probe; Tecnovet S.L., Centelles, Spain) was used to measure backfat thickness and loin depth on all pigs of 4 randomly selected Mp pens of each dietary treatment, 2 of each sex, representing a total of 251 pigs.

### 2.2. Feeding and Analysis

Two isoenergetic diets (2460 kcal NE/kg) based on maize, wheat, and soybean meal (Table 1) were formulated to meet or exceed all nutritional requirements, except lysine. Essential amino acids (AAs) were formulated based on the ideal protein ratios [35]. The low SID Lys:NE diet was 3.25 (Feed A) whereas the high one was 4.88 g SID Lys/Mcal NE (Feed B). Soybean meal inclusion was increased whereas maize inclusion was reduced to increase SID Lys:NE. In addition, the amount of crystalline AA in the diet was also modified. Feed was produced in successive blending batches (5000 kg). After pelleting, feed samples were collected for each blending batch and analyzed for crude protein (CP) (ISO 16634-2:2016) before used to ensure that levels were similar to the calculated. The 2 manufactured feeds were blended in 5 different proportions (Table 2) at the farm using a robotic feeding system to obtain the experimental treatments (DryExact Pro; Big Dutchman, Vechta, Germany). Furthermore, AA composition, by chromatography of hydrolyzed feed samples, and CP (Method 994.12) [36] were analyzed in a blend of the different batches of the 2 manufactured feeds (Table 3).

### 2.3. Calculations and Statistical Analyses

Body weight, ADG, average daily feed intake (ADFI), feed to gain ratio (F/G), SID Lys/kg gain and feed cost per kg gain were measured and calculated for the 3 phases (Phase 1: 28 to 46 kg BW – d 0 to 26, Phase 2: 46 to 63 kg BW – d 26 to 47, Overall: 28 to 63 kg – d 0 to 47) for each pen. In addition, metabolic ADFI was calculated by correcting ADFI with the metabolic BW (BW^0.6^) [37] at the middle of the phase to determine whether there were differences between BWCAT. Statistical analyses were carried out with R [38]. Models and ANOVA were performed using the *nlme* package [39], while contrasts and least square means were computed with the *emmeans* package [40]. The interactive effects between BWCAT and SID Lys:NE were analyzed in a linear mixed model using SID Lys:NE, BWCAT, their interaction and sex as fixed effects, while room was included as a random effect. Orthogonal polynomial contrasts for equally spaced treatments were implemented to evaluate if the linear or quadratic trends to increasing SID Lys:NE for each variable differed between BWCAT. In addition, orthogonal polynomial contrasts were implemented in the same model to each BWCAT. For all variables, a model with the triple interaction between SID Lys:NE, BWCAT and sex was tested, but as it was not significant for any variable, it was not included in the final model. Regarding ultrasound measures, a model without the BWCAT effect was built using the pig as experimental unit. It included SID Lys:NE, sex as fixed effects, individual BW at the end of the trial as a covariate, and room and pen within the room as random effects. Tables present least square means and standard errors computed with the *emmeans* package [40]. Results were considered significant when *p* ≤ 0.05 and tendency when 0.05 < *p* ≤ 0.10.

Moreover, the effects of SID Lys:NE on ADG and feed efficiency were modelled for the *Overall* period modifying the models outlined by Robbins et al. (2006) [41]. Fitted statistical models included were the broken-line linear ascending (BLL), broken-line quadratic ascending (BLQ) and quadratic polynomial (QP). Models were built with the *nlme* package of R [39], using room and BWCAT nested within the room as the grouping variables for ADG, whereas only BWCAT for feed efficiency as room did not improve the models fit. Models were fitted to predict ADG and gain to feed (G:F), which was preferred to F/G because enabled representing an ascending model. To improve the fitting process ADG was expressed in g and G:F in g/kg. As suggested by Pinheiro and Bates [42], only fixed effects parameters that accounted by the between-subject variability were left in the random effects formula. The inclusion was based on comparing the Bayesian information criterion (BIC). Besides, a weights statement was included in the G:F models to account for the linear increase in the variance along with the fitted values observed in the residual plots. After fitting, the 3 fitted models were compared using the BIC. Confidence intervals (CI, 95%) for the optimum SID Lys:NE to maximize the response were computed with the *nlme* package for BLL and BLQ. For QP models, the CI of the SID Lys:NE at which the response was maximized were estimated using the delta method in the *msm* package [43]. Additionally, models were fit for the *Overall* period for each BWCAT to compare the estimated SID Lys:NE requirements. Only one model was used for each comparison depending whether the response was linear or quadratic for all the BWCAT. All the models included room as a random effect, and the same procedure as for the general model was used to decide which fixed parameters were included in the random formula.

## 3. Results

The analyzed AA content of the experimental feeds was close to the calculated composition (Table 3). Only significant differences in CP for Feed A were reported, but initial analysis of the different blending batches reported a 14.8 ± 0.16% CP, always below the 0.5 error assumed by the method. Thus, the authors were confident that feeds adequately met the expected composition. Moreover, one Mp observation was removed from *Phase 2* and *Overall*, because from d 26–47 half of the pigs had an ADG ≤ 0.600 kg whereas the average ADG of the other 2 entire male pens of the same SID Lys:NE was 0.901 kg. Figure 1 summarizes the main effects of BWCAT on growth performance. Body weight was different between the 3 BWCAT throughout the experiment (*p* < 0.001), being 70.0, 62.7 and 55.6 kg at the end of the trial for Lp, Mp and Sp, respectively. Thus, the difference in BW between Lp and Sp increased from 8.7 to 14.4 kg, between day 0 and 47, as a result of a greater ADG (*p* < 0.001), and ADFI (*p* < 0.001) of Lp during the entire experiment. Although not reported in Figure 1, the effect of BWCAT on ADFI was still significant (*p* = 0.001) when the values were corrected by the metabolic body weight, being 0.157, 0.152 and 0.147 kg/kg BW^0.6^, for Lp, Mp and Sp, respectively. Finally, BWCAT had a significant effect on F/G (*p* < 0.001), with Sp being the most efficient (2.06, 2.00 and 1.96 for Lp, Mp and Sp, respectively).

### 3.1. Interactive Effects of SID Lys:NE between BW Categories

The differential effect of SID Lys:NE between BWCAT is reported as the pairwise comparison of the linear and quadratic trends in each BWCAT. In addition, the same functional forms are reported separately for each BWCAT. The triple interaction n between SID Lys:NE, sex and BWCAT was initially tested reporting a *p* > 0.100 for all the response variables analyzed. Additionally, the linear and quadratic effects of SID Lys:NE on growth performance were compared between entire males and females. Only in *Phase 2* there was a significant interaction in the linear effect between sexes on ADG, F:G, SID Lys/kg gain, and feed cost/kg gain, and consequently also the *Overall* response was different for some variables between sexes. The interaction resulted from only reporting a linear effect of SID Lys:NE on growth performance for entire males. Table 4 illustrates the interactive effects between BWCAT and SID Lys:NE on growth performance of *Phase 1* (28–46 kg). Increasing SID Lys:NE had a linear response (*p* < 0.010) on ADG in all BWCAT, but without a different effect between categories. Although increasing SID Lys:NE linearly reduced F/G in all BWCAT (*p* < 0.001), Sp pigs showed a greater linear reduction than Lp (*p* = 0.042). Similarly, SID Lysine used per kg gain increased more when increasing SID Lys:NE in Lp than in Sp (*p* = 0.005). Finally, the same interaction in the linear response to SID Lys:NE between Sp and Lp was also shown for feed cost per kg gain (*p* = 0.019), as it was linearly reduced in Sp whereas no significant linear effect was reported for Lp.

Interestingly, during *Phase 2* (46–63 kg) there were interactions between BWCAT and the linear effect of SID Lys:NE for almost all response variables except for ADFI, but not for the quadratic effect (Table 5). The linear increase in ADG as a response to increasing SID Lys:NE was greater for Mp (*p* = 0.011) and Sp (*p* = 0.028), both compared to Lp, for which there was no evidence of a linear effect (*p* = 0.217). Although there was no significant interaction on the effect on ADFI, Lp showed linear reduction (*p* = 0.006) and Mp a quadratic response (*p* = 0.016) to increasing SID Lys:NE. Regarding F/G, there was an interaction in the linear response to SID Lys:NE between Lp and Mp (*p* = 0.008) and Lp and Sp (*p* = 0.019). Whereas Lp did not show a significant linear reduction on F/G when increasing SID Lys:NE (*p* = 0.540), both Mp and Sp did (*p* < 0.001). Similarly, SID Lys intake per kg gain greatly linearly increased in Lp compared to Mp (*p* = 0.008) and Sp (*p* = 0.019). Finally, increasing SID Lys:NE had a substantial linear negative impact on Lp pigs feed cost per kg gain compared to both Mp (*p* = 0.006) and Sp (*p* = 0.017).

The interactive effects between BWCAT and SID Lys:NE on BW are presented in Table 6. Initially and at day 26, there was an interaction in the quadratic trend between Lp and Mp, explained by the slightly lower initial BW of Mp at 4.88 g SID Lys/Mcal NE, an unexpected effect from the randomization process. At day 26, increasing SID Lys:NE linearly increased BW in the 3 categories, and a quadratic response was shown by Mp. At the end of the trial, increasing SID Lys:NE increased linearly Sp BW (*p* < 0.001), whereas quadratically Mp (*p* = 0.001) and Lp (*p* = 0.041) BW. As there was no evidence of a linear effect on Lp (*p* = 0.183) BW, a tendency (*p* = 0.055) for a different linear response on final BW depending on SID Lys:NE between Lp and Sp was reported.

Results for the *Overall* period (Table 6) showed significant interactions between BWCAT and the linear effect of SID Lys:NE for all the response variables analyzed except for ADFI. Small pigs showed a greater linear increase (*p* = 0.030) in ADG when increasing SID Lys:NE than Lp whereas there was only a tendency for the same interaction between Mp and Lp (*p* = 0.093). A quadratic effect on Lp ADG was reported (*p* = 0.018) whereas for Mp and Sp both a linear (*p* < 0.001) and quadratic effect were reported (*p* = 0.002 and 0.047, for Mp and Sp, respectively). As in *Phase 1*, there was a tendency (*p* = 0.057) for a different quadratic response on ADFI between Mp and Sp. Increasing SID Lys:NE had a negative linear impact on Lp ADFI (*p* = 0.014), and a quadratic effect was reported for both Lp and Mp (*p* = 0.028 and 0.002, respectively). As expected from results in the other phases, Mp (*p* = 0.006) and Sp (*p* = 0.002) showed a greater linear reduction in F/G when increasing SID Lys:NE. Similarly, a significant interaction was also reported between Lp and Mp (*p* = 0.004) and Lp and Sp (*p* < 0.001) regarding the linear increase of SID Lys per kg gain. The reported value in the 3 BWCAT was numerically equal for the lower ratio (3.25 g/Mcal) but increased more in Lp compared to the Mp and Sp, both showing similar values across treatments. Finally, increasing SID Lys:NE showed a greater negative linear impact on feed cost per kg gain of Lp compared to Mp (*p* = 0.004) and to Sp (*p* = 0.001). Large pigs feed cost increased linearly (*p* < 0.001) whereas there was no evidence of an effect of SID Lys:NE on Mp, and a quadratic effect was reported for Sp (*p* = 0.007). What stands out in this table is that for all three SID Lys intake per kg gain, F/G and feed cost per kg gain the output for the lowest ratio was similar across BWCAT.

### 3.2. Effects of SID Lys:NE on Backfat Thickness and Loin Depth

A linear (*p* = 0.003) but not quadratic (*p* = 0.517) effect of increasing SID Lys:NE on backfat thickness of Mp at the end of the trial was observed (Figure 2). Backfat thickness was reduced from 6.25 to 5.50 mm comparing 3.25 and 4.88 g SID Lys/Mcal NE. Regarding loin depth, neither a linear (*p* = 0.261) or quadratic (*p* = 0.984) effect of SID Lys:NE were reported although it numerically increased from 47.8 to 49.3 mm between the lowest to highest SID Lys:NE tested. No evidence of an interaction (*p* ≥ 0.130) between SID Lys:NE and BW or sex was observed for backfat thickness.

### 3.3. Modelling the Response to SID Lys:NE

The best fitting BLL, BLQ and QP models to describe the effect of SID Lys:NE on ADG and G:F for the entire population from 28–63 kg BW are plotted in Figure 3 (see Appendix A for specific equations). Regarding ADG, each model provided different optimum SID Lys:NE to maximize performance. Those were 3.72 g/Mcal (95% CI: [3.58, 3.86]), 3.91 g/Mcal (95% CI: [3.55, 4.27]) and 4.40 g/Mcal (95% CI: [4.21, 4.59]) for BLL, BLQ and QP, respectively. In addition, maximum ADG was reported at 760, 755 and 770 g, and BIC was 911, 905 and 905 for BLL, BLQ and QP, respectively. Therefore, according to the BIC output BLQ and QP were the best fitting models, while BLL showed a slightly poorer fit. With respect to G:F, BLL reported the optimum at 4.29 g/Mcal (95% CI: [4.04, 4.53]), BLQ at 4.77 g/Mcal (95% CI: [4.27, >4.88]), and QP > 4.88 g/Mcal (95% CI: [4.25, >4.88]). As the optimum for the QP model was outside the range of the experiment, it was just considered to be greater than the maximum SID Lys:NE level. Nevertheless, when comparing the CI, the 3 models optimums were overlapped, with the lower boundary between 4.04–4.27 g/Mcal. Finally, BIC was 785 for the BLL, and 774 for both BLQ and QP models. Thus, both quadratic models fitted better the data although a lower CI was reported for the BLL model.

Having modeled the response for the overall population, models were fitted for each BWCAT with the aim to compare the requirements to optimize growth performance in each category. Results presented in Table 6 showed that there was no linear response for Lp pigs, and therefore fitting BLL models to that group was not possible. Consequently, requirements for ADG were compared using QP models, as a significant quadratic trend was reported for all BWCAT. The QP models (Figure 4) reported that Lp pigs maximized their ADG at 4.29 g/Mcal (95% CI: [3.91, 4.67]), Mp at 4.33 g/Mcal (95% CI: [4.13, 4.53]) and Sp at 4.60 g/Mcal (95% CI: [4.02, >4.88]) (see Appendix B for specific equations). As all the CI were overlapped, it was not possible to conclude whether the estimated requirements were different between categories. Although there were not differences in the optimum between Lp and Mp pigs, when comparing the requirements to reach 98% of the maximum ADG those were 3.67, 3.88 and 4.06 for Lp, Mp and Sp, respectively. This might be explained by the different marginal efficiency of increasing SID Lys:NE on each category, greater for Mp than for Lp.

Regarding feed efficiency, as there was a linear response in all BWCAT, BLL models were preferred to compare the requirements to maximize G:F. Large pigs maximized their feed efficiency at 4.29 g/Mcal (95% CI: [3.68, >4.88]), Mp at 4.75 g/Mcal (95% CI: [4.21, >4.88]) and Sp at 4.36 g/Mcal (95% CI: [4.00, 4.73]). Thus, there were no apparent differences in the requirement to maximize G:F, as the 3 CI were overlapped. Nevertheless, the plateau of maximum G:F was higher for Sp and Mp pigs, 536 g/kg (95% CI: [524, 548]) and 532 g/kg (95% CI: [517, 547]), respectively, compared to 499 g/kg (95% CI: [486, 512]) for Lp. Confirming, as indicated by the previous results comparing the linear and quadratic responses for each variable, that there is a greater range of improvement when increasing SID Lys:NE in Sp and Mp than in Lp.

## 4. Discussion

The widespread adoption of all-in all-out swine production systems has raised concerns on the issue of pigs with a low BW at marketing [1,2,3]. Those pigs, which are already smaller at the nursery exit [2] make phase feeding strategies inefficient and inaccurate [5,14]. The large amount of literature available on this topic suggest that this is a multi-factorial problem, which cannot be confronted in a single strategy. The 9 kg BW difference between Sp and Lp when starting the experiment was greater than most available studies [17,24,29], but other works have even reported initial greater differences [2]. Because of the great difference, it might be impossible for Sp to reach the same BW as Lp at a specific marketing time. Furthermore, a lightweight at the end of the nursery has been related with reduced ADG during the grow-finishing [2], but some degree of compensatory growth might be expected by those pigs if sufficient nutrients are provided [44]. Thus, worthwhile strategies to avoid increasing the differences in BW between Lp and Sp pigs might focus on maximizing ADG of low BW pigs [28,45].

Most studies have just focused on the effect of increasing AA and/or energy on low BW pigs using a single nutritional strategy [17,29]. Consequently, no similar experiment was found in the literature regarding the divergent effect of increasing SID Lys:NE on pigs classified in different BWCAT. In this study, isoenergetic but not isoproteic diets were preferred to ensure that essential AA were not limiting, while avoiding that a great proportion of non-essential AA were in excess in the low SID Lys:NE diets, which would have to be deaminated. The differences in ADFI have been suggested as one of the main drivers for the reduced growth of low BW pigs. The reported 15% lower ADG and a 19% lower ADFI of Sp compared to Lp, was in agreement with several studies focusing mainly on the nursery stage [9,10,11,12]. Douglas et al. [29] reported a similar reduction on ADG (17%) but contrarily to this study, it was a result of a 16% increase in F/G as no differences in ADFI were reported. Other studies suggested that when ADFI was corrected by the metabolic BW, there were no evidences of differences [9,10]. Although in this study the differences were reduced when expressed by metabolic BW, Sp had a 6% significantly lower feed intake, similar to the 4% lower energy intake reported by van Erp et al. [11]. Future works might aim to answer whether Sp, with a reduced feed intake, might respond also to greater energy densities or if just SID Lys intake limits their performance.

Phase feeding commercially widespread strategies consist in a feeding program in which the amounts that must be fed of each feed are decided focusing on the average pig, although a 10% security margin has been suggested by some authors [14]. Considering the results from *Phase 1*, the differences in ADFI would suppose that in 26 days Sp would eat 7.8 kg less feed and considering a 1% SID Lys feed, 78 g SID Lys less than Lp, which would be a reduction of a 20% compared to Lp. Consequently, when using phase feeding strategies those pigs might be limited in SID Lys available for growth compared to their heavier mates. López-Vergé et al. [17] showed that a strategy to provide the same amount of the initial grower feed to small pigs as the amount fed to the average population improved ADG during the grow-finishing period. However, as the diets were only tested on low BW pigs, doubts arise whether some response would have been observed on the heavier pigs. Although studies involving different phase-feeding strategies might give an indication if growth of small pigs is impaired or improved by different strategies, these effects have not been reported in those works [7,8].

### 4.1. Effect of BWCAT on the Response to SID Lys:NE

In this study, a different linear effect of SID Lys:NE was reported between pigs classified in the 3 BWCAT. Unexpectedly, only for *Phase 2* the response on ADG between Sp and Lp differed. It might be explained because SID Lys:NE did not limit growth performance of Lp throughout the second phase. Therefore, the statistical model was more powerful when comparing a BWCAT in which there was no linear effect with one with a linear effect, than when there was a linear effect in both BWCAT. Focusing on a range in which the Lp category might not show a response on growth performance whereas Sp might, would be a strategy for future works using a similar experimental design. The low F/G reached by Mp at the highest SID Lys:NE was considered by the authors the result of a lower ADFI rather than an improved performance. Thus, as expected, *Overall* results confirmed that the effect of SID Lys:NE was different between Lp and Sp for all growth performance variables studied, except for ADFI.

The greater response of Sp pigs to SID Lys:NE might be related to several factors, and therefore those will be further discussed. Although only some Mp were ultrasound measured, previous results have confirmed that last pigs harvested are leaner than the first ones [16,19]. This might support the idea that the ratio between energy deposited as protein or lipids is greater in slow growing pigs [22], and therefore energy was more efficiently used for growth [46]. For instance, Sp used feed more efficiently for growth only when pigs were allowed to eat a high SID Lys:NE diet. Therefore, as isocaloric diets were used, probably Sp were more efficient because a greater fraction of energy was being used for protein deposition [46]. Similarly to the present study, Main et al. [47] reported only a reduction of fat depth on 35–60 kg gilts when increasing SID Lys:NE, but not in longissimus muscle area. Further works might aim to determine if feeding higher SID Lys:NE to Sp and Mp increases the differences in carcass fatness observed without nutritional interventions [16,18,19].

The different linear response of entire males and females during *Phase 2* could be expected as entire males are known to have a greater potential for protein deposition [48]. Thus, this study provided evidence of a different response between sexes to increasing SID Lys:NE starting around 50 kg. Although this trial with only pigs from one sex might have reduced the chance of confounding effects, the authors considered that employing both sexes gave a better indication of the expected outcomes in real commercial production systems. In addition, as the inclusion of the SID Lys:NE and sex interaction did not modify the conclusions from the results presented, the simplest model was preferred. Summarizing, in the context of current all-in all-out swine commercial production systems, more research is needed to corroborate the results presented, and confirm that split feeding pigs by BWCAT can improve the overall population performance and reduce associated costs.

### 4.2. Critical Assessment of SID Lys:NE Requirement Models

The inconsistencies between nutrient requirement models have been underlined by many authors previously [41,49,50] and confirmed in the present study. Although some studies just published the best fitting model [51], considering that model choice depends also on how nutritional requirements and marginal responses are understood [49], we decided to publish the 3 different models. Besides comparing model fit, some authors have also mentioned the relevance of the CI, particularly when a break point is included in the model [49]. As expected, the BLL yielded the lowest requirement [50], but although showing the narrowest CI of the requirement, it fitted worst the observations. Reported requirements to maximize growth performance would be greater than NRC [6], which were on average 3.70 g SID Lys/Mcal NE from 25–75 kg BW. Only BLL requirements for ADG would be similar to NRC, although that requirement was the average between ADG and feed efficiency. Recent studies have reported lower requirements for G:F, being around 4.5 g SID Lys/Mcal NE from 25–50 kg [52,53] and 3.6 from 50–75 kg [52]. However, this might be affected by the different response for ADG and G:F reported in this study.

As nutritionists, we would expect from this study a different requirement for each BWCAT, which enables deciding whether Sp pigs should be fed 10 or 20% higher SID Lys:NE compared to Lp. However, the dose-response models fitted to each BWCAT suggest careful consideration in accordance with the lack of evidence of a different quadratic response. The QP models showed that Sp pigs maximized ADG at a higher SID Lys:NE, but the CI were overlapped for the 3 BWCAT. Nevertheless, the different linear response might suggest a reduced diminishing marginal productivity [49] for those pigs. Regarding G:F, although requirements were not significantly different, the models confirmed the greater potential of Sp to efficiently use high SID Lys:NE diets. The wide CI reported indicates that a considerable larger number of replicates [49] would be necessary to determine different requirements between BWCAT using the conventional dose-response modelling approaches.

Moreover, Goodband et al. [54] suggested that although there are changes in the SID Lys:NE requirements along with genetic improvements, nursery pigs maximize their growth performance at around 19 g SID Lys intake per kg gain across different studies. Recent studies in growing pigs have also shown that growth performance between 30–60 kg is maximized between 19–21 g/kg [47,53]. If we consider that the same hypothesis applies to different BWCAT, then excess lysine above 19–20 g/kg might not be used for protein deposition and consequently deaminated [55]. In the present study, it might be assumed that *Overall* ADG was numerically maximized at 3.66, 4.07 and 4.47 g SID Lys/Mcal NE, with an efficiency of use of SID Lys per kg of 18.8, 19.7 and 20.6 g SID Lys/kg gain for Lp, Mp, and Sp, respectively. Therefore, with those results, it might be impossible to conclude that pigs in different BWCAT maximize ADG at the same SID Lys per kg gain. Other studies have shown that increasing SID Lys reduces its efficiency of utilization [56], which might be related to changes in maintenance requirements or alterations of AA composition. Future studies might analyze the differences in lysine needed for maintenance and the differences in protein composition of pigs in different BWCAT. Nevertheless, independently of the requirement models for each BWCAT, the results presented confirmed that Sp and Mp pigs had a greater potential to improve their growth performance when fed increasing SID Lys:NE.

## 5. Conclusions

This work confirmed that a different response to SID Lys:NE can be expected from growing pigs (28–63 kg BW) sorted in different initial BW categories. However, the traditional models to estimate nutrient requirements failed to give significant different results for each category. Thus, an applied perspective of these results might be based on the different diminishing marginal productivity of small pigs compared to large pigs when increasing SID Lys:NE. In addition, the results of modelling the general population showed that SID Lys:NE requirements might be greater than NRC, especially for maximizing gain to feed ratio. Important practical implications are that feeding pigs sorted by initial BW different SID Lys:NE during the growing phase might be feasible to maximize performance of small and medium pigs and reduce costs of large pigs.

## Figures and Tables

**Figure 1 animals-10-01032-f001:**
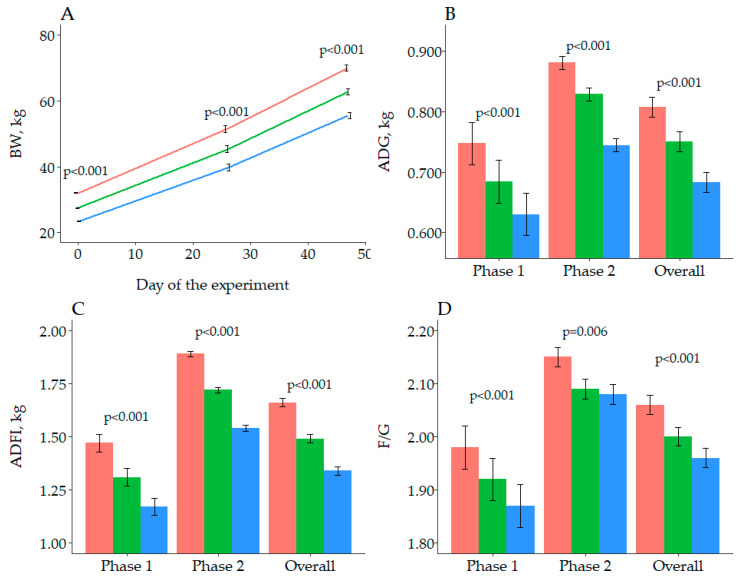
Effects of initial body weight category (Large/Lp: red; Medium/Mp: green; Small/Sp: blue) on body weight (**A**), average daily gain (**B**), average daily feed intake (**C**) and feed to gain (**D**) of growing pigs in Experiment 1. Results in A are provided for the 3 weighing days. For B, C and D results are provided for *Phase 1* (d 0–26), *Phase 2* (d 26–47) and *Overall* (d 0–47). Error bars represent the standard error of the mean.

**Figure 2 animals-10-01032-f002:**
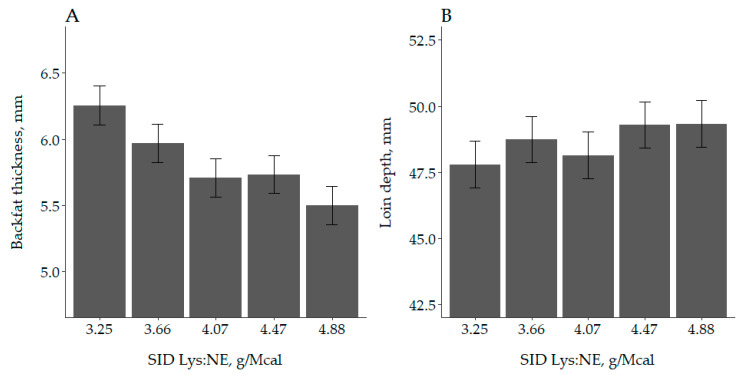
Effect of standardized ileal digestible lysine to net energy ratio (SID Lys:NE) on final backfat thickness and loin depth of medium body weight category pigs measured using an ultrasound scan at P2. (**A**) Backfat thickness was linearly (*p* = 0.004) reduced when increasing SID Lys:NE whereas there was no evidence of an effect of SID Lys:NE on (**B**) loin depth (*p* = 0.261).

**Figure 3 animals-10-01032-f003:**
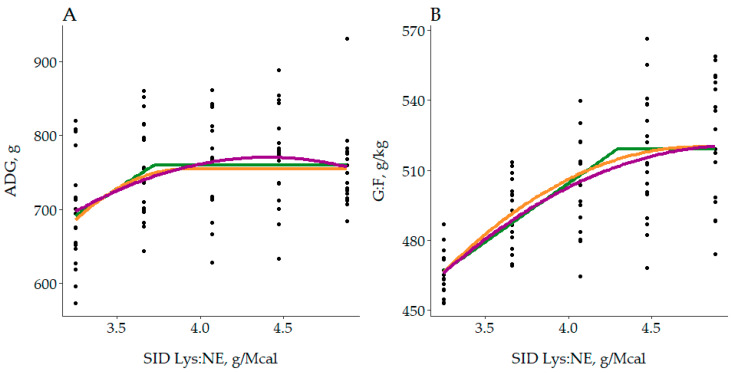
Fitted broken-line linear (BLL, green), broken-line quadratic (BLQ, orange) and quadratic polynomial (QP, magenta) regressions models to optimize (**A**) average daily gain (ADG) and (**B**) gain to feed (G:F) as a function of standardized ileal digestible lysine to net energy ratio (SID Lys:NE) from 28–63 kg (Overall phases). For ADG, the BLL model estimated the optimum at 3.72 g/Mcal (95% CI: [3.58, 3.86], BIC = 908), the BQL at 3.91 g/Mcal (95% CI: [3.55, 4.27], BIC = 905) and the QP at 4.40 g/Mcal (95% CI: [4.21, 4.59], BIC = 905). Regarding G:F, BLL estimated the optimum at 4.29 g/Mcal (95% CI: [4.04, 4.53], BIC = 785), BLQ at 4.77 g/Mcal (95% CI: [4.27, >4.88], BIC = 774) and QP > 4.88 g/Mcal (95% CI: [4.25, >4.88], BIC = 774), both outside the range of the experiment.

**Figure 4 animals-10-01032-f004:**
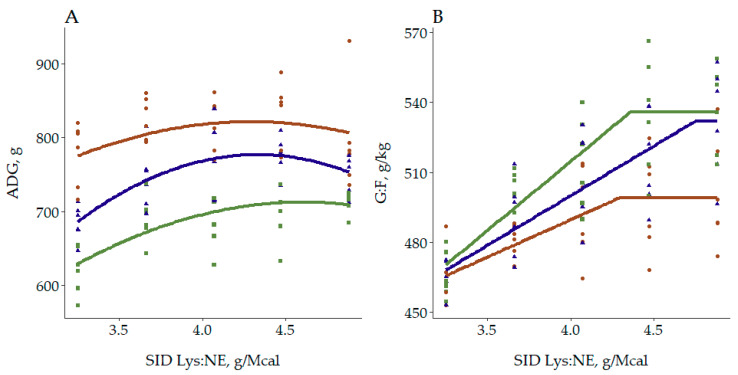
Fitted regressions models to optimize (**A**) average daily gain (ADG) using quadratic polynomial models (QP) and (**B**) gain to feed (G:F) using broken-line linear (BLL) models as a function of standardized ileal digestible lysine to net energy ratio (SID Lys:NE) from 28–63 kg (Overall phases) for each body weight category (Large/Lp: brown, Medium/Mp: blue and Small/Sp: green) in Exp 1. For ADG, the QP models estimated the optimum of Lp at 4.28 g/Mcal (95% CI: [3.91, 4.67]), of Mp at 4.33 g/Mcal (95% CI: [4.13, 4.53]), and of Sp at 4.60 g/Mcal (95% CI: [4.02, >4.88]). Regarding G:F the BLL models estimated the optimum of Lp at 4.29 g/Mcal (95% CI: [3.68, >4.88]), of Mp at 4.75 g/Mcal (95% CI: [4.21, >4.88]), and of Sp at 4.36 g/Mcal (95% CI: [4.00, 4.73]).

**Table 1 animals-10-01032-t001:** Ingredient and calculated composition (as fed basis) of the feeds used for blending the 5 dietary treatments.

Ingredient Composition, %	A	B	Calculated Composition ^1^	A	B
Maize	45.36	38.35	Dry matter, %	87.56	87.84
Wheat	35.00	35.00	Crude Fiber, %	2.74	2.80
Soybean meal	13.70	19.20	Sugars, %	2.70	2.96
Choice white grease	1.90	2.30	Starch, %	48.90	44.42
Calcium carbonate	1.16	1.18	Ether extract, %	4.19	4.46
Dicalcium phosphate	0.70	0.61	Crude Protein, %	14.49	17.42
Sodium chloride	0.40	0.41	Total Lysine, %	0.89	1.30
Lysine sulphate	0.52	1.03	SID Lysine, %	0.80	1.20
L-Threonine	0.14	0.34	SID Met+ Cys/Lys ratio	0.60	0.60
DL-Methionine	0.07	0.27	SID Thr/Lys ratio	0.68	0.68
L-Valine	-	0.15	SID Trp/Lys ratio	0.20	0.20
L-Tryptophan	0.02	0.08	SID Val/Lys ratio	0.68	0.65
L-Isoleucine	-	0.06	SID Ile/Lys ratio	0.60	0.53
Phytase ^2^	0.02	0.02	ME, kcal/kg	3266	3287
Liquid Acid mix ^3^	0.30	0.30	NE, kcal/kg	2460	2460
Solid Acid mix ^4^	0.40	0.40	SID Lys:NE, g/Mcal	3.25	4.88
Premix VIT-MIN ^5^	0.30	0.30	Ashes, %	4.17	4.39
			Total Ca, %	0.68	0.68
			Total P, %	0.44	0.44
			STTD P, %	0.36	0.36
			Cl, %	0.28	0.28
			K, %	0.56	0.64
			Na, %	0.16	0.17

^1^ SID: standardized ileal digestible; ME: metabolizable energy; NE: net energy; STTD: standardized total tract digestible. ^2^ 6-phytase (750 FTU/kg). ^3^ Blend of formic and lactic acid. ^4^ Blend of medium chain fatty acids. ^5^ Provided per each kg of feed: 4500 IU vitamin A, 2000 MIU vitamin D_3_, 15 mg vitamin E, 0.7 mg vitamin K, 1.0 mg vitamin B_1_, 4.0 vitamin B_2_, 1.2 vitamin B_6_, 0.02 vitamin B_12_, 15 mg niacin, 12 mg pantothenic acid, 107 mg of choline from choline chloride, 90 mg Fe from iron sulphate, 100 mg Zn from zinc sulphate, 50 mg Mn from manganese oxide, 20 mg Cu from copper sulphate, 1.8 mg I from potassium iodide and 0.25 mg Se from sodium selenite.

**Table 2 animals-10-01032-t002:** Blend ratios, calculated composition, and calculated price of the 5 dietary treatments (as-fed basis) averaged for the inclusion of each feed.

Item	SID Lys:NE g/Mcal ^1^				
	3.25	3.66	4.07	4.47	4.88
Feed A, %	100	75	50	25	0
Feed B, %	0	25	50	75	100
SID Lys, %	0.80	0.90	1.00	1.10	1.20
NE, kcal/kg	2460	2460	2460	2460	2460
Cost, €/tn ^2^	242.1	251.8	261.6	271.3	281.0

^1^ Calculated standardized ileal digestible lysine to net energy ratio. ^2^ Formula cost.

**Table 3 animals-10-01032-t003:** Analyzed (A) versus calculated (C) AA composition (%, as fed basis) of the feeds used for blending the 5 dietary treatments.

Item	Feed A		Feed B	
	A	C	A	C
Crude Protein	15.00	14.49	17.31	17.42
Lys	0.92	0.89	1.27	1.30
Met	0.27	0.28	0.48	0.50
Cys	0.26	0.27	0.28	0.30
Met + Cys	0.53	0.55	0.76	0.80
Thr	0.63	0.62	0.89	0.90
Val	0.67	0.64	0.86	0.88
Arg	0.57	0.55	0.69	0.71
His	0.37	0.36	0.40	0.42
Ile	0.57	0.55	0.69	0.71
Leu	1.17	1.16	1.24	1.30

**Table 4 animals-10-01032-t004:** Interactive effects between initial body weight category (BWCAT) and standardized ileal digestible lysine to net energy ratio (SID Lys:NE) on growth performance on *Phase 1* (d 0–26).

Item ^3^	BWCAT ^4^	SID Lys:NE, g/Mcal					SEM ^5^	*p*-Value			
		3.25	3.66	4.07	4.47	4.88		L × BW ^1^	Q × BW ^1^	Linear ^2^	Quad. ^2^
ADG, kg	Large	0.691	0.738	0.764	0.788	0.757	0.0389	-	Mp/Sp ^†^	0.003	0.027
	Medium	0.622	0.669	0.732	0.713	0.684				0.006	0.001
	Small	0.565	0.631	0.628	0.652	0.676				<0.001	0.401
ADFI, kg	Large	1.46	1.52	1.48	1.50	1.41	0.048	-	Mp/Sp ^†^	0.228	0.035
	Medium	1.30	1.33	1.36	1.32	1.23				0.094	0.007
	Small	1.17	1.21	1.16	1.16	1.17				0.427	0.916
F/G	Large	2.12	2.05	1.94	1.91	1.88	0.046	Lp/Sp *	-	<0.001	0.114
	Medium	2.10	1.99	1.86	1.86	1.80				<0.001	0.011
	Small	2.08	1.93	1.85	1.77	1.73				<0.001	0.021
Lys/gain, g/kg	Large	17.0	18.5	19.4	21.0	22.6	0.47	Lp/Sp **	-	<0.001	0.413
	Medium	16.8	17.9	18.6	20.4	21.6				<0.001	0.153
	Small	16.6	17.3	18.5	19.5	20.7				<0.001	0.319
Cost/gain, €/kg	Large	0.513	0.517	0.508	0.518	0.528	0.0121	Lp/Sp *	-	0.139	0.193
	Medium	0.509	0.502	0.486	0.503	0.507				0.882	0.033
	Small	0.503	0.485	0.483	0.481	0.485				0.063	0.068

Least square means. ^1^ Pairwise comparison of the linear (L × BW) or quadratic (Q × BW) effect of SID Lys:NE between BWCAT: ^†^ 0.05 < *p* ≤ 0.10, * ≤0.05, ** ≤0.01. ^2^ Orthogonal linear or quadratic (Quad.) contrasts on the effects of SID Lys:NE on each BWCAT. ^3^ ADG: average daily gain; ADFI: average daily feed intake; F/G: feed to gain; Cost/gain: feed cost per kg gain. ^4^ BWCAT: initial body weight category of the pen was large (Lp, 32.1 ± 2.8 kg), medium (Mp, 27.5 ± 2.3 kg) or small (Sp, 23.4 ± 2.9 kg). ^5^ SEM: standard error of the mean.

**Table 5 animals-10-01032-t005:** Interactive effects between initial body weight category (BWCAT) and standardized ileal digestible lysine to net energy ratio (SID Lys:NE) on growth performance on *Phase 2* (d 26–47).

Item ^3^	BWCAT ^4^	SID Lys:NE, g/Mcal					SEM ^5^	*p*-Value			
		3.25	3.66	4.07	4.47	4.88		L × BW ^1^	Q × BW ^1^	Linear ^2^	Quad. ^2^
ADG, kg	Large	0.874	0.923	0.873	0.873	0.856	0.0225	Lp/Mp * Lp/Sp *	-	0.217	0.319
	Medium	0.766	0.847	0.827	0.855	0.846				0.018	0.124
	Small	0.686	0.764	0.751	0.763	0.754				0.057	0.072
ADFI, kg	Large	1.91	1.96	1.88	1.88	1.82	0.029	-	-	0.006	0.217
	Medium	1.69	1.77	1.74	1.75	1.66				0.340	0.016
	Small	1.53	1.58	1.56	1.52	1.53				0.490	0.415
F/G	Large	2.19	2.13	2.16	2.16	2.13	0.038	Lp/Mp ** Lp/Sp *	-	0.540	0.717
	Medium	2.22	2.09	2.11	2.05	1.97				<0.001	0.860
	Small	2.24	2.06	2.07	2.00	2.03				<0.001	0.024
Lys/gain, g/kg	Large	17.5	19.1	21.6	23.7	25.6	0.40	Lp/Mp ** Lp/Sp *	-	<0.001	0.877
	Medium	17.7	18.8	21.0	22.5	23.7				<0.001	0.654
	Small	17.9	18.6	20.7	22.0	24.4				<0.001	0.097
Cost/gain, €/kg	Large	0.530	0.535	0.564	0.585	0.600	0.0101	Lp/Mp ** Lp/Sp *	-	<0.001	0.777
	Medium	0.537	0.526	0.551	0.556	0.554				0.043	0.950
	Small	0.541	0.519	0.543	0.542	0.571				0.011	0.042

Least square means. ^1^ Pairwise comparison of the linear (L × BW) or quadratic (Q × BW) effect of SID Lys:NE between BWCAT: * ≤0.05, ** ≤0.01. ^2^ Orthogonal linear or quadratic (Quad.) contrasts on the effects of SID Lys:NE on each BWCAT. ^3^ ADG: average daily gain; ADFI: average daily feed intake; F/G: feed to gain; Cost/gain: feed cost per kg gain. ^4^ BWCAT: initial body weight category of the pen was large (L, 32.1 ± 2.8 kg), medium (M, 27.5 ± 2.3 kg) or small (S, 23.4 ± 2.9 kg). ^5^ SEM: standard error of the mean. Different SEM for Medium at 4.07 g SID Lys/Mcal NE (1 observation removed). Values were 0.0246, 0.032, 0.042, 0.43 and 0.0110 for ADG, ADFI, F:G, Lys/gain and Cost/gain, respectively.

**Table 6 animals-10-01032-t006:** Interactive effects between initial body weight category (BWCAT) and standardized ileal digestible lysine to net energy ratio (SID Lys:NE) on body weight (BW) and growth performance on the *Overall* experiment (d 0–47).

Item ^3^	BWCAT ^4^	SID Lys:NE, g/Mcal					SEM ^5^	*p*-Value			
		3.25	3.66	4.07	4.47	4.88		L × BW ^1^	Q × BW ^1^	Linear ^2^	Quad. ^2^
BW d 0	Large	32.0	32.2	31.8	32.2	32.1	0.22	-	Lp/Mp *	0.597	0.679
	Medium	27.3	27.6	27.7	27.7	27.1				0.596	0.015
	Small	23.5	23.3	23.6	23.4	23.3				0.778	0.796
BW d 26	Large	50.0	51.3	51.7	52.7	51.8	1.15	-	Mp/Sp *	0.006	0.073
	Medium	43.5	45.0	46.7	46.2	44.9				0.031	<0.001
	Small	38.1	39.7	40.0	40.4	41.1				<0.001	0.475
BW d 47	Large	68.3	70.8	70.0	71.1	69.7	1.12	Lp/Sp ^†^	-	0.183	0.041
	Medium	59.7	62.9	63.9	64.2	62.8				0.002	0.001
	Small	52.7	55.7	55.9	56.7	57.0				<0.001	0.073
ADG, kg	Large	0.773	0.822	0.813	0.826	0.800	0.0210	Lp/Mp ^†^ Lp/Sp *	-	0.195	0.018
	Medium	0.689	0.750	0.773	0.777	0.759				<0.001	0.002
	Small	0.621	0.690	0.687	0.706	0.712				<0.001	0.047
ADFI, kg	Large	1.66	1.72	1.66	1.67	1.59	0.029	-	Mp/Sp ^†^	0.014	0.028
	Medium	1.48	1.52	1.52	1.51	1.42				0.096	0.002
	Small	1.33	1.38	1.34	1.32	1.33				0.349	0.591
F/G	Large	2.15	2.08	2.04	2.02	2.00	0.027	Lp/Mp ** Lp/Sp **	-	<0.001	0.172
	Medium	2.16	2.04	1.97	1.94	1.88				<0.001	0.065
	Small	2.15	1.99	1.95	1.87	1.86				<0.001	0.002
Lys/gain, g/kg	Large	17.2	18.8	20.4	22.3	24.0	0.29	L/M ** L/S ***	-	<0.001	0.565
	Medium	17.2	18.3	19.7	21.4	22.5				<0.001	0.795
	Small	17.2	17.9	19.5	20.6	22.4				<0.001	0.053
Cost/gain, €/kg	Large	0.520	0.525	0.532	0.548	0.561	0.0071	L/M ** L/S ***	-	<0.001	0.266
	Medium	0.522	0.513	0.514	0.527	0.528				0.149	0.166
	Small	0.520	0.500	0.509	0.509	0.524				0.417	0.007

Least square means. ^1^ Pairwise comparison of the linear (L × BW) or quadratic (Q × BW) effect of SID Lys:NE between BWCAT: ^†^ 0.05 < *p* ≤ 0.10, * ≤0.05, ** ≤0.01. ^2^ Orthogonal linear or quadratic (Quad.) contrasts on the effects of SID Lys:NE on each BWCAT. ^3^ ADG: average daily gain; ADFI: average daily feed intake; F/G: feed to gain; Cost/gain: feed cost per kg gain. ^4^ BWCAT: initial body weight category of the pen was large (L, 32.1 ± 2.8 kg), medium (M, 27.5 ± 2.3 kg) or small (S, 23.4 ± 2.9 kg). ^5^ SEM: standard error of the mean. Different SEM for Medium at 4.07 g SID Lys/Mcal NE (1 observation removed). Values were 0.0219, 0.030, 0.028, 0.31 and 0.0075 for ADG, ADFI, F:G, Lys/gain and Cost/gain, respectively.

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
