# Peer review of "Increasing Dietary Lysine Impacts Differently Growth Performance of Growing Pigs Sorted by Body Weight"

_animals, 2020, doi:10.3390/ani10061032_

Round 1
Reviewer 1 Report
This is a good work, that was meticulously designed and it is finely presented. I have no comments on my evaluation and I believe it is ready for publication in this form. Congratulations to the authors. It is an important piece of work for animal nutritionist, especially those involved in precission pig feeding.
Reviewer 2 Report
Animals 814134 review “Increasing dietary lysine impacts differently growth 3 performance of growing pigs sorted by body weight)
This is a very interesting study looking at the effect of liveweight on lysine requirements that has been well written. The data will be a great interest to commercial pork producers. However, the discussion (and perhaps the results) is very long for a manuscript that is really only reporting on growth performance albeit some interesting aspects.
line 24 – first sentence of the abstract doesn’t make sense.
line 27 – Need to define SID Lys:NE. Even though defined in Simple summary it needs to also be defined when it first appears in the abstract as well as the full body of the manuscript.
line 30 – ..fed over 47 days.
line 34 and elsewhere – maybe one of the reasons for the wide confidence interval is that the authors chose 98% rather than 95% of maximum growth rate. Using 98% results in very high estimates of requirements and because the curve is on a plateau increases the confidence interval.
Line 34 – overlapping
line 42 – ..as they have been associated…
line 48 – Lighter pigs also have a higher relative maintenance and protein deposition rates (relative to liveweight and energy intake) which will increase lysine requirements.
Line 55 – I would not say that this study has been refuted as many others have also reported similar findings that light for age pigs are fatter. It is more that there are contradictory findings in the literature.
Line 57 – ..so straight forward…
Line 63 – …as they considered…
Line 132 and elsewhere – p needs to be italicised with a space before and after the sign.
Line 163-164 – I am not sure what this sentence about blending means. What does 14.8 % refer to. I assume it means that the first blend of Diet A had a CP of 14.8% which was close to expected (15%) and this gave you confidence to continue. If this the case I am fine with it but just need to be explicit.
Figure 1 – Why not report FCE in Figure 1 instead of FCR to be consistent with the curve fitting.
Line 216 – …during the experiment.
Figure 2 and elsewhere – Backfat thickness and loin eye depth are not measures of body composition but rather linear measures that some people try to relate to body composition with limited success. Just refer to them as what they are.
Discussion – the discussion is very long and needs to be condensed substantially. While this is very good work it doesn’t warrant such a lengthy discussion as the authors only really measured growth performance and no mechanisms.
Reviewer 3 Report
Comments to Author
The submitted manuscript reports a study investigating the effects of dietary lysine on growth performance for growing pigs with different body weight. The results showed that increasing dietary lysine was beneficial to the growth of low-weight pigs, but excessively increasing Lys levels has no positive effect on medium-weight to large-weight pigs. Overall, layout of the manuscript is robust and the main assumptions are put forward on a scientifically solid basis. However, the following points should be addressed:
- Although the author declares that a full-in and full-out production mode is adopted in the experiment, it is still necessary to describe the initial age of each group, which affects the accuracy of results.
- The author achieved a linear increase in the ratio of SID lysine/NE through the mixing of two isoenergetic diets. There are obvious differences in the amount of protein ingredient in the two diets, which resulted in a big difference between the CP content and AA composition. Although the addition of crystalline amino acids maintains the ratio of essential AA to maintain the ideal protein pattern, the content or ratio of non-essential AA that can affect pig metabolism is not clear. Will this increase the error of the test results? Therefore, the author needs to answer why he did not choose to directly add L-Lys to adjust Lys/NE. What are the advantages of the current method?
- The author measured the SID of two diets according to the method of AOAC 994.12, but the process of this method still needs to be briefly described.
- The authors compared the effect of Lys/NE ratio on backfat thickness and loin depth in Mp-pigs, but it did not involved whether different BW change the effect of Lys/NE ratio on the carcass composition of pigs, so it is necessary to increase the description for this section in the results or discussion.
Round 2
Reviewer 3 Report
All comments have been addressed adequately. The manuscript can be recommended for acceptance.